# Boots on the Ground and Eyes in the Sky: A Perspective on Estimating Fire Danger from Soil Moisture Content

**Sonisa Sharma** *,† and **Kundan Dhakal** †

Noble Research Institute, LLC, 2510 Sam Noble Parkway, Ardmore, OK 73401, USA; kdhakal@noble.org
* Correspondence: sonisa@okstate.edu
† These authors contributed equally to this work.

**Abstract:** With increasing forest and grassland wildfire trends strongly correlated to anthropogenic climate change, assessing wildfire danger is vital to reduce catastrophic human, economic, and environmental loss. From this viewpoint, the authors discuss various approaches deployed to evaluate wildfire danger, from in-situ observations to satellite-based fire prediction systems. Lately, the merit of soil moisture in predicting fuel moisture content and the likelihood of wildfire occurrence has been widely realized. Harmonized soil moisture measurement initiatives via state-of-the-art soil moisture networks have facilitated the use of soil moisture information in developing innovative applications for wildfire prediction and risk management applications. Additionally, the increasing availability of remote-sensing data has enabled the monitoring and modeling of wildfires across various terrestrial ecosystems. When coupled with remotely sensed data, field-based soil moisture measurements have been more valuable predictors of assessing wildfire than alone. However, sensors capable of acquiring higher spectral information and radiometry across large spatiotemporal domains are still lacking. The automation aspect of such extensive data from remote-sensing and field data is needed to rapidly assess wildfire and mitigation of wildfire-related damage at operational scales.

**Keywords:** wildfire; live fuel moisture content; soil moisture; remote sensing

## 1. Background

Fire is an inevitable and essential ecological process in many fire-dependent terrestrial ecosystems of the biosphere [1]. Fire plays an intricate role in shaping the landscape and population structure and composition of inhabiting species in those specific fire-dependent ecosystems. Such ecosystems require fire for maintaining ecosystem functioning, including regulations of water and energy cycles, fuel accumulations, and removal of pests and pathogens [2,3]. Anthropogenic activity has led to a disturbance of the delicate balance between fire activity and the natural ecosystem regeneration process, resulting in increased wildfire activity, especially in western North America, Australia, Canada, Greece, Portugal, and France [4,5]. In fire science literature, fire danger has often been described as both constant (topography and fuel) and variable (wind, fuel moisture, and fuel temperature) determinants that affect the initiation, spread, and difficulty of controlling a wildfire within a specific area. Fire danger is used for a broad-scale assessment and forecast of potential fire, often expressed as low-to-extreme rating levels. Based on the rating level, fire managers, first respondents, and residents can better understand the relative seriousness of possible fire and develop their plan and recommended actions for each category level.

The consequences of unplanned wildfires result in increased emissions of air pollutants, greenhouse gas (GHG), and particulate matter; surface albedo; runoff; soil degradation; desertification; and reduced evapotranspiration, thereby affecting carbon budgets in the burned landscapes [6,7]. Fire suppression has reduced the global total burned area since the 1930s [8], with a decline of 18% since 2000 [9]. However, annual global fire, including grassland fire, forest fire, crown fire, peat fire, and agricultural residue burning [10],

is estimated to affect three to five million square kilometers, resulting in the release of $\approx 2.2$ Pg C yr$^{-1}$ into the atmosphere [11]. Satellite observation of the earth's surface depicts an average of 4.63 Mkm$^2$ burned globally [12]. In 2019–2020 alone, the Amazon rainforest witnessed a massive wildfire, burning 20,234 km$^2$ [13]. In recent decades, the U.S. has experienced several large, medium, and small wildfires across the forest, grassland [14], and urban areas, resulting in firefighter and civilian fatalities, livestock loss, and structural damages [15–17]. Likewise, in Canada, 8300 forest fires have occurred over 25 years, burning an average of 23,000 km$^2$ per year [18]. In the western U.S., a significant increasing trend has been observed for fire season length, the number of large fires, and the annual burned area [4,5]. Unfortunately, rising wildfires within the wildland–urban interface and associated losses have also been reported across savannas of Africa and forests of Europe, South America, and Asia [19–21].

Because of the economic, environmental, and human loss associated with small- and large-scale wildfires globally, there is an urgency to monitor wildfire impacts and develop tools to prepare wildfire mitigation, response, and recovery. This article reviews multiple tools and techniques currently deployed for wildfire preparedness and the outlook of wildfire management plans. In the following discussion, we aim to present our observations in our understanding of wildfire, emerging tools and techniques (Figure 1), and the outlook—mainly from the North American perspective and a grassland wildfire focus; however, examples from other regions and ecosystems are also discussed wherever relevant.

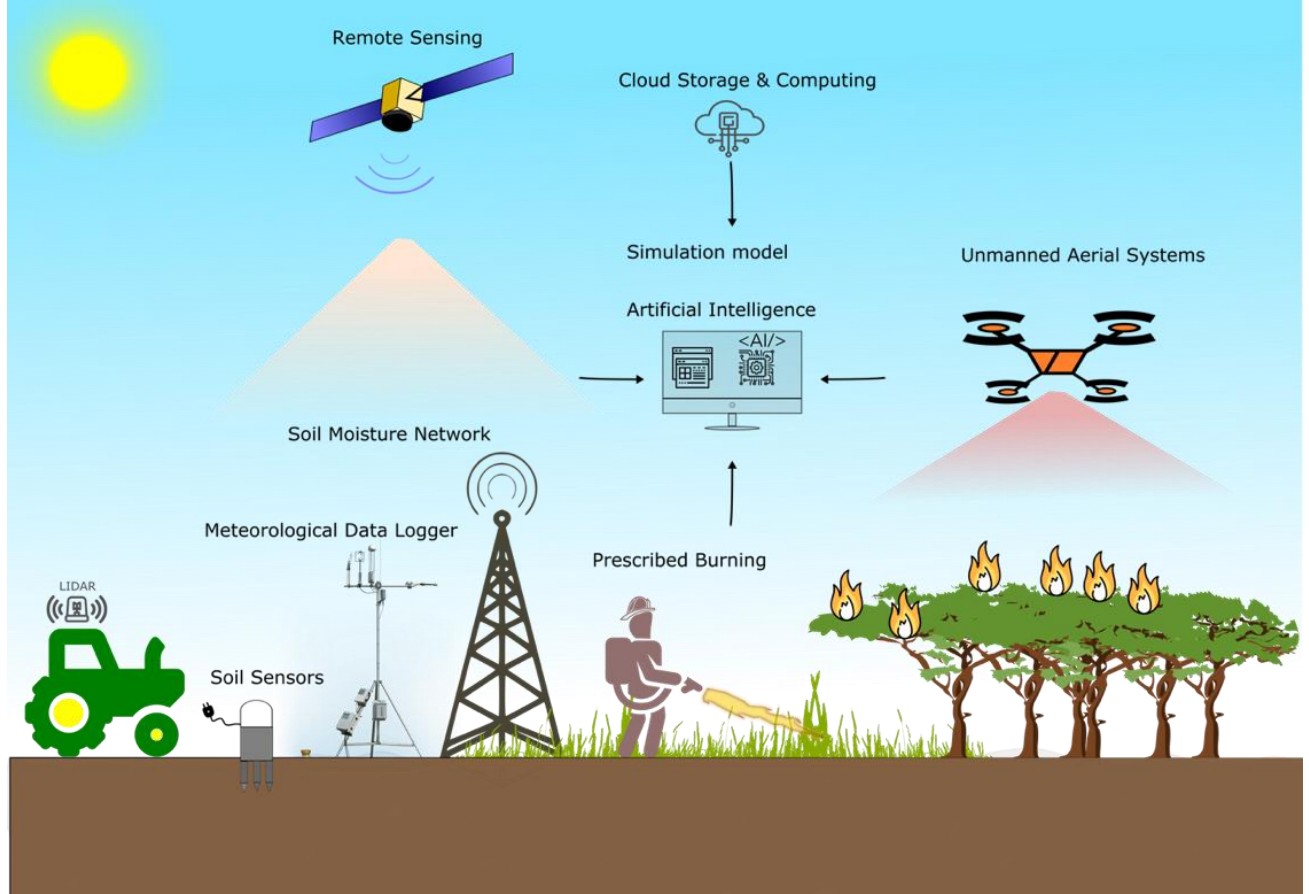

**Figure 1.** Illustration depicting how current wildfire prediction and analytics are data driven. Data range from soil sensors installed in a plot, information on fire management practices at the field level, uncrewed aerial systems, acquired hyperspectral data, LiDAR data, and remotely sensed satellite information at the landscape level. Quality data obtained at each scale are critical for wildfire prediction analytics and fire simulation studies.

## 1.1. Fire Danger Appraisals

Various attempts have been made to document and develop tools to track wildfire danger. There are different fire danger rating systems used across the world, such as the European Forest Fire Information System (EFFIS), Canadian Wildland Fire Information System, U.S. National Fire Danger Rating System (NFDRS), Australian McArthur's Fire Danger Rating System, Russian Nesterov Index, and Malaysian and Indonesian Fire Danger Rating System.

The U.S. NFDRS was developed in the early 1970s. The NFDRS is a complex operational system and uses the relationship between various fuels (live and dead), meteorological variables, topography, and vulnerability. The NFDRS has multiple inputs such as ignition component (probability of a fire requiring suppression), spread component (forward rate of spread of a headfire), energy release component (composite fuel moisture value or the available energy per unit area within the flaming front at the head of a fire), burning index (numeric value closely related to the flame length in feet times 10), lightning occurrence index (relationship between lightning activity and ignition component), human-caused fire occurrence index (derived from the relation between human activity and the fire start potential), fire load index (maximum effort required to contain all possible fires within a specific area and time), and Keetch–Byram drought index (measures the seasonal drought effect on fire potential).

To provide a "one-stop" repository of live and dead fuel moisture records, the National Fuel Moisture Database was created in 2006. The data are administered by personnel of each of the Geographic Area Coordination Centers. Before its establishment, such daily data were available in various formats, which was challenging to synthesize coherently. Currently, information on the FMC is available from the Wildland Fire Assessment System (WFAS) (*WFAS*. https://www.wfas.net, accessed on 21 July 2021). The daily FMC data are available as a gridded netCDF file for the entire continental United States. Lately, Quan et al. [22] developed the first daily global FMC data at 500 m resolution from 2001–2019 using the Moderate Resolution Imaging Spectroradiometer (MODIS) and radiative transfer models (TMs). Before that, most of the FMC data were only available for specific fire-prone regions of Canada, the U.S., Australia, and Spain.

In Canada, research on forest fire danger rating was commenced by J.G. Wright in 1925. Later, during the 1960s, Forestry Canada led the development of the Canadian Forest Fire Danger Rating System (CFFDRS). The CFFDRS has various outputs, such as fire behavior, active burning fires, fire weather normal, and monthly and seasonal forecasts. It provides the foundation for understanding the fire environment and obtaining early warning of potential wildfire events [23]. The CFFDRS, with its subsystem Fire Weather Index (FWI) and its intermediate components, the Burning Index (BI) and its components of the NFRDS, the components of Mark 5, and the Keetch–Byram Drought Index, uses meteorological data such as maximum temperatures (dry bulb temperature) and total precipitation of the previous day [24].

Similarly, Australia uses the McArthur Fire Danger Rating Systems (FDRS), which is a significant component of the Forest Fire Danger Index (FFDI), also called Mark 5 [25]. In Europe, the European Forest Fire Information System (EFFIS), part of the Copernicus Emergency Management Service, is a modular decision support system that monitors forest fires at a continental scale [26,27]. The FFDI uses relative humidity, dry bulb temperature, wind speed, and dryness of the soil. The FWI also uses temperature, relative humidity, wind speed, precipitation, and day length. The burning index's input is weather information and fuel model as well as the slope of the land [28].

## 1.2. Fuel Moisture, Live Fuel Moisture Content, and Dead Fuel Moisture Content

Some commonly used terminologies in the wildfire community for assessing wildfire danger, viz. fuel moisture content, live fuel, dead fuel, live fuel moisture content, and dead fuel moisture content, are discussed here. Fuel moisture content is the water content present in the fuel, expressed as a percent of the oven-dry weight of the fuel, which can

be described either as live or dead fuels. Live fuels are naturally occurring fuels whose moisture content is controlled by physiological processes within the plant. Live fuels are grouped as herbaceous annual, herbaceous perennial, or woody. Live fuel moisture content (LFMC) is defined as the mass of water per unit dry biomass in vegetation that exerts direct control on fuel ignitability, fuel availability, and fire spread. Hence, the LFMC is an essential parameter in wildfire risk assessment. Dead fuels are categorized according to their time lag, e.g., 1, 10, 100, and 1000 h. The moisture content of the dead fuel is affected by precipitation, temperature, and relative humidity.

### 1.3. Meteorological and Topographic Variables on Fire Behavior Potential

Most fire danger rating systems rely on meteorological data such as air temperature, relative humidity, wind speed, wind velocity, and precipitation in studying the initiation, spread, and tracking of fires by several wildfire danger rating systems, such as OK-FIRE [29], U.S. National Fire Danger Rating systems (NFDRS) [30], and the Fosberg Fire Weather Index (FFWI) [17]. Seager et al. [31] presented vapor pressure deficit as a helpful indicator of vegetation water stress, depicting a relationship with burned areas across the southwest U.S. Near-real-time swath data of land-surface temperature with a temporal resolution of 5 min derived from Terra Moderate Resolution Imaging Spectroradiometer (MODIS) in Land, Atmosphere Near real-time Capability for Earth Observing System of the National Aeronautics and Space Administration was found useful for forecasting forest fires in four quadrants around the globe [32]. Moreover, monthly vapor pressure deficit, soil moisture data, and the Global Fire Emission Database when masking out fires on agricultural land were used to predict fire danger for each geographic region in the U.S. [33].

Land use and topography also play a vital role in wildfire ignition and spread [34]. Dalezios [35] reported slope, fuel type, and fuel moisture content as essential indicators for wildfire rating. Notably, the slope was found to be the most prominent topographical element to model fire occurrence in Alberta, Canada [18]. Regardless, fuel moisture content (FMC), the water ratio in the live and dead biomass related to dry biomass, has been widely recognized as a critical variable influencing wildfire ignition and spread [36–38].

### 1.4. Boots on the Ground: Field-Based Sampling and Monitoring

Fuel moisture content is the most critical variable for ignition and fire spread [39]. LFMC has been found to be helpful in inferring wildfire occurrence and wildfire behavior around the globe [40]. Field-based sampling has been the go-to method of estimating LFMC and deriving empirical relationships for predicting wildfires. The LFMC is obtained through the gravimetric method. The gravimetric process is incredibly time-consuming and expensive. As discussed earlier, meteorological forcing information, including temperature, rainfall, vegetation, and vapor pressure deficit, has proven to be helpful in predicting wildfire activity [31]. LFMC can be more strongly related to soil moisture in some cases than remote-sensing measurements [40].

Field-measured LFMC was significantly related to microwave soil moisture data 60 days prior over the conterminous U.S. when using time-lagged robust linear regression models [41]. In addition, field-measured soil moisture and the normalized difference vegetation index (NDVI) derived from a multispectral radiometer inferred a curing rate (the rate of transition of live to dead fuel, at soil moisture value; volumetric, unitless) <0.36 in tallgrass prairie in Oklahoma, U.S. [20]. Gabriel et al. [42] used 21 years of field LFMC data in the Mediterranean region of Catalonia to validate post- and pre-effects of fire. Such data are expected to help in wildfire risk assessments and the validation of remote-sensing products [43].

### 1.5. Soil Moisture as a Proxy for LFMC

In the large area of Oklahoma, soil moisture was used when LFMC measurements were not available [44]. The study reported that field-based soil moisture was more critical

and sensitive to large growing-season wildfires than the Keetch–Byram Drought Index (KBDI), an index used to determine fire potential [44]. Various other studies reported that soil moisture information has proven to infer wildfire danger at regional, national, and global scales [45–48]. LFMC was strongly and significantly related to soil moisture 60 days prior to LFMC sampling [40]. Soil moisture (0–35 cm) was found to be a significant contributor for predicting LFMC with a correlation of 0.74 in Australia using remote-sensing-based LFMC and land-surface-based soil moisture [49]. The reason behind the use of soil moisture in inferring wildfire danger is the complex relationship between soil moisture and other environmental factors such as temperature, wind, vapor pressure deficit (VPD), and rainfall. As there is a decrease in soil moisture, high evaporation leads to wildfire risk because of environmental factors such as high temperature, high wind, high VPD, and low rainfall. Evaporative demand drains soil moisture, and this can be used as a proxy to determine wildfire potential in advance. In addition, the response of evaporative demand in the form of evapotranspiration is more in peatland than in forests with warming-induced VPD, as suggested by Helbig et al. [50]. Based on the study of 1907 fire ignition points in the western U.S. from July to August of 2015–2018, the author found that remotely sensed soil moisture from the Soil Moisture Active Passive satellite, along with VPD, allowed for improved predictive skill in wildfire modeling [51].

Fuel moisture estimation from field-based sampling, though accurate, may not be relevant when studying on a broader scale due to underlying spatial variation across landscapes. The same reasoning can apply to the FMC received from automated weather stations that are often distributed sparsely, leading to increased uncertainty, bias, and error [52].

## 1.6. Eyes in the Sky: Remote-Sensing Tools and Applications

The advantage of satellite-based soil moisture products and various soil moisture networks provide more opportunities to use soil moisture at the field and satellite scales to estimate fuel moisture and curing rate, which are particularly important for assessing wildfires [45]. The high correlation with the water absorption spectrum on spectral reflectance over a larger spatial footprint and the variation of LFMC, which is strong enough to discriminate among soil, vegetation, atmosphere, sensor geometry, and plant characteristics, enable remote-sensing implementation at scales commensurate with regional wildfire risk assessment [48]. Since the early 1970s, remote-sensing products have been successfully used to inform us of the availability and abundance of fuel and fuel type and perform pre-, during-, and post-fire analyses [53].

In the visible and infrared spectra, the National Oceanic and Atmospheric Administration (NOAA-N), geostationary Metostat, and Geostationary Operational Environmental Satellite (GOES), and environmental satellites such as Landsat, Spot, Worldview, European Remote Sensing (ERS-1), Japanese Earth Resources Satellite (JERS-1), Moderate Resolution Imaging Spectroradiometer (MODIS), and Sentinel are typical in wildfire impact assessment. There are various indices developed from remotely sensed imagery that have been widely used to study wildfires, and readers are suggested to refer to the review article by Chowdhury and Hasan [3] for an in-depth discussion of these various indices used in fire danger monitoring.

In this section, the authors highlight some of the remote-sensing applications in wildfire danger assessment. Satellite-based Soil Moisture and Ocean Salinity (SMOS) derived that soil moisture is a significant predictor for wildfire danger assessment in the Iberian Peninsula [54]. In the fire-prone region in the Iberian Peninsula, LFMC was estimated from the empirical models developed from NOAA-AVHRR-derived NDVI, surface temperature, and day of the year. The LFMC data one week before fire detection was found to be a crucial factor in determining ignition probability [55]. However, this study and other studies could not find the exact threshold of LFMC for different ecosystems [45,55–57].

Similarly, soil moisture derived from the Soil Moisture Active Passive L-band radiometer (SMAP) and cumulative growing degree days were found to be good predictors to

estimate LFMC in the Mediterranean ecosystem of Southern California, U.S. [41]. Simulated satellite soil moisture from NASA's Gravity Recovery and Climate Experiment (GRACE) and the historical fire data from the USDA Forest Service in the U.S. from 2003–2012 suggested that the GRACE's simulated soil moisture correlated with wildfire activity [47]. Like Rothermel's fire-spread model, the fire-behavior model uses LFMC as a proxy estimate from weather-based indices such as the Keetch–Byram Drought Index in satellite remote sensing [58]. The red edge position developed from Vegetation and Environmental New Micro Spacecraft (VENµS), a satellite explicitly designed for Mediterranean species, was found to be better at estimating LFMC than the shortwave infrared (SWIR) band developed from MODIS and Sentinel-2 data despite the lack of a shortwave infrared band in VENµS [59]. The authors attributed that to the high spatial heterogeneity in the Mediterranean vegetation, which was better captured by the VENµS's 10-m spatial resolution, and also suggested the use of deep learning algorithms to combine long-term remotely sensed information with field sampling that captures various vegetation types along with their phenological stages [60]. In addition, microwave remote sensing has been helpful for assessing plant water content [59] and for estimating daily LFMC because the microwave has a longer wavelength and low sensitivity to atmospheric and cloud effects [61]. In this study, it was found that microwave root zone soil moisture was better at capturing the variability of LFMC than near-surface soil moisture. The MODIS-derived Enhanced Vegetation Index (EVI2) and NDVI estimated live fuel moisture content in non-native tropical grasslands in Hawaii with $R^2$ = 0.46, which was better than the National Fire Danger Rating System ($R^2$ = 0.37) and KBDI ($R^2$ = 0.06) [62].

In-situ measured records of live fuel moisture content from 11 countries from 1977–2018 have been used to create a global LFMC database [63]. Likewise, LFMC was estimated using microwave backscatter Sentinel-1 and optical reflectance Landsat-8 and then validated using field data from 125 sites from the National Fuel Moisture Database [64]. The data were trained using a deep learning model with an accuracy of $R^2$ = 0.69 in shrublands [64]. Similar results were found when field LFMC data were validated ($R^2$ = 0.72 to 0.75) against estimated LFMC using empirical models in Sentinel-2 and MODIS images in *Cistus ladanifer,* a fire-prone species in Mediterranean areas [65].

Lately, with the development and improvement of sensors, hyperspectral information for studying various landscape processes is on the rise. However, the cost associated with acquiring data with a fine spatiotemporal resolution (<10 m), mostly airborne hyperspectral data, may limit its use in developing empirical models for wildfire monitoring and prediction applications [66]. Even with the higher resolution sensors in QuickBird and Ikonos, quantifying the fuel underneath the canopy and in cloudy conditions is difficult [10]. However, with an active sensor such as Light Detection and Ranging (LiDAR), computing canopy metrics such as canopy height, canopy structure, crown bulk, etc., are attainable, which is helpful for estimating forest ladder fuel (fuel allowing vertical continuity for a surface fire into tree crowns) [67].

Though satellite observations provide considerable capabilities for assessing fire danger conditions, there are several shortcomings of using such information, as reviewed in depth by Yebra et al. [48]. For example, the disparity between ground observation and lower spectral resolution of MODIS was reported by Adelabu et al. [68]. Currently, there is a trade-off between coarse-scale and fine-scale spatiotemporal remote-sensed data that are used in wildfire assessment. Data with fine temporal resolution have a limitation with spatial resolution and vice versa. Satellite data with high to moderate spatial resolution are required for upscaling field-based measurements before coarse resolution data can be used to estimate FMC on a finer scale [68].

There is a need to calibrate remotely sensed soil moisture according to ecosystem type, topography, etc., at a smaller scale to gain a better resolution. Once calibrated, remotely sensing soil moisture offers a better solution to gather more accurate conditions of FMC over a broader temporal and spatial scale. In addition, as slope affects the angle of incident electromagnetic radiation, the reliability of the soil moisture using Soil Moisture

and Ocean Salinity (SMOS) can be questionable, particularly in mountainous regions [69]. Unfortunately, the reflectance value might not be able to capture the variability of LFMC over time because of the time lag for capturing satellite data, as evidenced by the empirical and radiative transfer models derived for Moderate Resolution Imaging Spectroradiometer (MODIS) [34] and Landsat [46].

Early evidence that soil moisture, both field- and satellite-based, can assess wildfire danger assessment indicates that soil moisture measurements could be further exploited in the future and used to develop risk-based wildfire management. Predicting wildfire occurrence in the form of live fuel moisture content, fuel moisture content of mix at a broader scale will be a valuable tool to mitigate the loss associated with wildfires on a regional, national, and more comprehensive scale. Highlighting that live fuel moisture content at the field level is seldom absolute, the authors note that collecting field data will help infer wildfire danger assessment.

### 1.7. Artificial Intelligence and Machine Learning: State of the Art and the Way Forward

Information on the FMC available from the Wildland Fire Assessment System (WFAS) resulting from interpolation of the sparse automated weather station measurements from often sparsely distributed weather stations could lead to erroneous estimations of FMC. To address this issue, the National Center for Atmospheric Research (NCAR) implemented machine learning techniques with MODIS TERRA and AQUA data to generate daily one-kilometer gridded FMC data for the continental U.S., which can be retrieved as netCDF files from the NCAR's webpage [69]. However, with TERRA and AQUA being decommissioned, alternative platforms need to be developed to acquire similar information. Low-cost computation, access to high-performance cloud computing, and availability of planetary-scale satellite imagery have favored the use of novel machine learning algorithms in wildfire assessment. As the machine learning paradigm is being shaped, techniques such as random forest, artificial neural network, decision tree, and support vector machine have opened new avenues to understanding wildfires and wildfire danger [70]. Using machine learning and high-resolution dynamic vegetation maps and weather data, Smith et al. [71] generated a fine-scale rangeland fire forecast for the upcoming fire season for the Great Basin of the U.S. For an in-depth review of the use of machine learning applications in wildfire management, readers are suggested to refer to Jain et al. [70], where the authors discuss the use of the machine learning fuel characterization and mapping, fire susceptibility, fire behavior, fire effects, fire management, and fire weather and climate change.

To facilitate data access and processing, Climate Engine, a web-based application that integrates Google's Earth Engine framework to query, process, and display output in real time, provides the ability for the user to request four sets of fire danger indices computed from the gridMET data [72]. Rangeland Analysis Platform (RAP), an interactive web application, has been developed as a free tool for the public, and uses machine learning and Google Earth Engine to track vegetation and rangeland productivity over time. Developed by the Interagency Fuel Treatment Decision Support System [73], the RAP tracks rangeland vegetation for the western U.S., and requires spatially explicit information describing production, fuel, grazing capacity, and successional trajectory (https://rangeland.app; accessed on 23 July 2021). Likewise, a hybrid modeling approach coupled an adaptive neuro-fuzzy inference system with various metaheuristic optimization algorithms to classify landscapes with multiple levels of wildfire probability [74]. In most wildfire danger models, the spatial dependence of observations is not accounted for. Compared to commonly used global logistics modes, a geographically weighted logistic model with adaptive Gaussian spatial kernels was found to be effective in fire-presence modeling [75].

Nevertheless, the emerging capabilities of advanced machine learning algorithm implementation call for more ground-truth data to train these novel models. However, the success of machine learning models depends on the quality of training data available.

Hence, efforts need to be concerted in collating quality ground measurements across various regions of the world.

## 2. Outlook and Conclusions

There is plenty of research with promising results, including remote-sensing indices, a weather-based index, and soil moisture from both fields and satellites. Despite the remarkable advances in tools and technology, the current state of wildfire monitoring and risk assessment is still primitive relative to that required for accurate and large-scale information that has been approved with robust and ample ground-truth data. In the case of automated weather stations, their sparse siting and the lack of soil moisture sensors installed at each weather station have been a bottleneck for deriving various metrics and monitoring appropriate spatiotemporal resolution. High-resolution (<2.5 km) information is critical for accurate modeling of fire behavior. Proper soil moisture sensor depth assessment needs to be performed to improve wildfire models, as optimum soil depth sensors are highly dependent on soil type, root zone depth, time of year, and geography. More importantly, wildfire danger assessments that rely on LFMC should be cautious when using meteorological data, as they are vary depending on the site location and the plant community dynamics [38,57,76]. Overall, we must have a better understanding of climate dynamics, especially on rising temperature, changes in precipitation and intensity, and their interactions on the fire regime for proactive planning. Some areas of emphasis could be general circulation models and regional circulation models to estimate spatially explicit, detailed wildfire occurrence scenarios and provide changes in wildfire-impacted places compared to the baseline to fire management agencies [77]. This is possible by coupling macroscale fuel moisture monitoring with climatic modeling and remote sensing [78].

This paper focuses on using live vegetation and soil moisture as metrics for assessing fire danger. With enhanced satellite sensor technology, computational capacity (storage, cloud computing, machine learning, and image processing algorithms), along with a high-density weather monitoring network (with soil moisture and temperature sensors), informed fire danger assessments and improved preparedness are possible. Usage of soil moisture in fire-spread models may enable better prediction of LFMC, which in turn will be helpful for fire danger assessment, lowering the LFMC sampling cost. Likewise, identifying region-specific soil moisture thresholds could play a significant role in forecasting wildfires in the future.

For accurate fire danger assessment, soil moisture sensors should be installed in the most fire-prone regions. Low cost and easy-to-install soil moisture sensors and IoT could be an important breakthrough tool to assess wildfire danger on a local level. However, on landscapes with diverse plant community structures, high-resolution remotely sensed data are required to capture the soil moisture adequately and generate accurate estimates of LFMC. Current and accurate ground-truth data at an appropriate spatiotemporal resolution must be commensurate with the requirement for wildfire risk modeling and assessment efforts.

In the future, wildfire-related studies should involve gathering pertinent and high-quality ground observation data for different regions worldwide and integrate different fine spatiotemporal resolutions of relevant geospatial data. A paradigm shift is imperative to address the wildfire issue collectively across the various geopolitical and socio-ecological gradients. Detailed exploration is warranted, especially given the contemporary context of climate change, to mitigate future disasters effectively. Additionally, exploring the impact of drought on wildfire potential requires more attention and funding. Future work should investigate developing a universal model that can be applied to assess wildfire risk potential across landscapes and various ecosystem types to mitigate future wildfire disasters effectively. Lack of proper attention in wildfire studies will have pronounced pressure on our livelihood, ecosystem, and climate system.

**Author Contributions:** S.S. and K.D. fully contributed to all aspects of this article. Both authors have read and agreed to the published version of the manuscript.

**Funding:** This research received no external funding.

**Acknowledgments:** The authors thank Alistair M. S. Smith, and four anonymous reviewers for their valuable comments and constructive feedback. Their critical review of the manuscript and suggestions helped with substantial improvements.

**Conflicts of Interest:** The authors declare no conflict of interest.

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
