# Peer review of "Boots on the Ground and Eyes in the Sky: A Perspective on Estimating Fire Danger from Soil Moisture Content"

_fire, doi:10.3390/fire4030045_

Round 1

Reviewer 1 Report

See attached file

Author Response

Thank you so much.

Reviewer 2 Report

Line 55: What is “outlook of fire management plans”?

Line 62-64: What is the relevance of this statement for the paper?

Line 120: What is “238”?

Line 129: Define “risk”. Perhaps “potential” is a better term for the context of this statement.

Line 182: “Span” should be “spanned”.

Line 199: “are” should be “have been”.

Line 279: What does “infer decision-making” mean. Do you mean “inform decision-making”. And why just on large fires?

Line 287: Since “wildfire danger” has a specific meaning, perhaps this should be generic and just “wildfire”.

Line 288: This should “researched”, nor “research”, but most compared to what?

Author Response

Thank you.

Reviewer 3 Report

The manuscript discusses past efforts and more recent advancements of wildfire monitoring and assessment of wildfire risk using both ground- and satellite-based techniques. In my opinion the manuscript fits the scope of the journal and contributes to an overall understanding of the state-of-the-art in wildfire monitoring and assessment.

I suggest the authors better explain what is the importance of their work and why does the scientific community need a discussion/perspective article such as this one.

I was a bit confused by the overall organization of the article. You might want to reconsider what belongs to each section of the article. A better explanation of these sections at the end of the background section would be helpful. 

I've included more specific comments in the attached PDF document. I suggest this manuscript can be published after these concerns are addressed. 

Reviewer 4 Report

The paper, ‘Boots on the ground and eyes in the sky: A perspective on estimating fire danger’ by Sharma et al., attempts to give a perspective of current ‘on-the-ground’ and remote sensing techniques for estimating fire danger.

Overall, this paper is unfocused, unorganized and has numerous mistakes – see comments below. I think this could be a useful perspective, but the manuscript will need an overhaul to be publishable.

  1. This paper is unfocused and hard to follow.

The introduction should define what fire danger is and why it is important to quantify. Instead, it is a mishmash of fire statistics that make it very confusing to the reader what the paper is about.

There’s substantial text devoted to assessing fire severity (lines 173-194), which has very little to do with assessing fire danger. It’s very confusing why this text is in the paper.

There is almost no mention of using ground/RS techniques for estimating fuel models – a large component of assessing fire danger - despite ample examples in the literature (see Gale et al. 2021, RSE).

  1. Organization of the paper also makes it impossible for the reader to follow in some sections.

There’s a substantial chunk of RS text in the ‘Boots on the ground’ section (lines 131-149).

Many paragraphs start out on one topic and end on a completely different topic. For example, the paragraph at lines 214-234 starts out discussing RS of LFMC, the middle discusses using RS for measuring fuels and the last sentence talks about RS of fire severity.

  1. I noticed several mistakes, noted below. There are likely other mistakes given the obvious nature of these (you only need to look at the abstracts of these papers to know that they are cited incorrectly).

Line 47: Wildfire activity is actually decreasing in all of these listed areas (Andela et al. 2017, Science).

Lines 187-188: This sentence is incorrect, NBR was sensitive to soil type. Not sure what is meant by ‘effects of weather’ but that is not in the cited paper either.

Lines 188-190: This sentence is incorrect – MIRBI was calculated using Landsat ETM data, not LiDAR.

Round 2

Reviewer 4 Report

Overall, the manuscript has improved, and the authors have focused and organized the text in a better way. However, there are a few issues identified in the comments below that need to be addressed.

The ‘background’ section should define what fire danger is and why it is important to quantify. Instead, it is a mishmash of fire statistics that make it very confusing to the reader what the paper is about. Many readers with very little background in fire danger may read this, so it is critical to define what fire danger is, what are its usual components, and why we should care about it (i.e. why is quantifying fire danger useful?) in the first few paragraphs of the opening section.

Lines 273-278: If you’re not going to explain how these spectral indices could be useful in assessing wildfire danger, there’s no reason to list them – delete this text.

Lines 298-300: This paragraph is about estimating soil moisture from RS – why is there a sentence on estimating fuel load in the middle of it?

Conclusions:

What does “spatiotemporal resolution commensurate with management” mean?

This study focuses on using live vegetation and soil moisture as metrics for assessing fire danger, but neither are mentioned in the conclusions. What is the best way to assess these metrics? What are the positives and negatives of using these metrics? Where are the gaps that future studies should examine? – sum this up in the conclusions!

Figure 1: This figure isn’t really that useful, it’s just a collection of the paper’s topics in pictorial form. Perhaps a more useful figure could highlight previous work on quantifying live fuel and soil moisture using RS? It could include maps highlighting the spatial resolution of available datasets, and graphs highlighting the relationships between RS metrics and live vegetation/soil moisture.

Author Response

Thank you so much for your feedback.

Round 3

Reviewer 4 Report

Overall, the manuscript has improved, but there are a few issues identified in the comments below that need to be addressed.

I would suggest moving the ‘Fuel moisture, live fuel moisture content, and dead fuel moisture content’ section below the ‘Fire danger appraisals’ section. It makes more sense to talk about historical approaches to estimating fire danger first and then move into the components, e.g. fuel moisture, weather and topographical features.

Lines 158-162: Why is text on a post-fire BAER product in this paper? The focus of this paper is fire danger, not fire effects.

Line 279: What is a ‘fair’ spatiotemporal resolution?

Lines 283-287: It is distracting to the reader to have text on forest structure/fuels in a section mostly focused on remote sensing of soil/fuel moisture.

Line 299: Don’t start a paragraph with ‘Besides’

Lines 350-360: It isn’t clear why text on the Rangeland Vegetation Simulator is in the ‘Artificial intelligence and machine learning section’. Either clarify that it uses AI or ML or delete it from this section.

Lines 368-369: Stating that wildfires occur every year globally and that they are highly researched is obvious – please delete or rephrase this sentence.

Line 370: Again, why is burn severity mentioned in a fire danger paper?

Author Response

Thank you.
